# Consumption of Antibiotics and Epidemiology of *Clostridioides difficile* in the European Union in 2016—Opportunity for Practical Application of Aggregate ECDC Data

**DOI:** 10.3390/antibiotics9030127

**Published:** 2020-03-19

**Authors:** Estera Jachowicz, Anna Różańska, Monika Pobiega, Mariusz Topolski, Jadwiga Wójkowska-Mach

**Affiliations:** 1Department of Microbiology, Faculty of Medicine, Jagiellonian University Collegium Medicum, 31-121 Kraków, Poland; estera.jachowicz@doctoral.uj.edu.pl (E.J.); a.rozanska@uj.edu.pl (A.R.); monika.pobiega@gmail.com (M.P.); 2Department of Systems and Computer Networks, Wroclaw University of Science and Technology, 50-370 Wroclaw, Poland; mariusz.topolski@pwr.edu.pl

**Keywords:** *Clostridioides difficile*, healthcare-associated infections, antibiotic consumption, statistical methods of data classification

## Abstract

Background: The most important pathomechanism of *Clostridioides difficile* infections (CDI) is post-antibiotic intestinal dysbiosis. CDI affects both ambulatory and hospital patients. Aim: The objective of the study was to analyze the possibility of utilizing databases from the European Centre for Disease Prevention and Control subject to surveillance for the purpose of identifying areas that require intervention with respect to public health. Methods: The analysis encompassed data concerning CDI incidence and antibiotic consumption expressed as defined daily doses (DDD) and quality indicators for antimicrobial-consumption involving both ambulatory and hospital patients in 2016. Results: In 2016, in the European Union countries, total antibiotic consumption in hospital and outpatient treatment amounted to 20.4 DDD (SD 7.89, range 11.04–39.69); in ambulatory treatment using average of ten times more antibiotics than hospitals. In total, 44.9% of antibiotics used in outpatient procedures were broad-spectrum antibiotics. We have found a significant relationship between the quality of antibiotics and their consumption: The more broad-spectrum antibiotics prescribed, the higher the sales of antibiotics both in the community sector and in total. CDI incidence did not statistically significantly correlate with the remaining factors analyzed on a country-wide level. Conclusion: Antibiotic consumption and the CDI incidence may depend on many national variables associated with local systems of healthcare organization and financing. Their interpretation in international comparisons does not give clear-cut answers and requires caution.

## 1. Introduction

*Clostridioides difficile* is the most significant pathogen responsible for antibiotic-related diarrhea, and the most critical exposure factor is the consumption of antibiotics that disrupt the gut microbiota [1]. The antibiotics associated with the highest risk of development of *Clostridioides difficile* infections (CDI) are clindamycin, beta-lactams and fluoroquinolones [2]. CDI may affect both hospital and ambulatory patients. However, it is not easy to distinguish CA-CDI (community-associated CDI) from CDI acquired in the hospital (HA-CDI, healthcare-associated CDI). If symptoms of CDI appear after 48 h from admission to the hospital or in the period of four weeks after discharge from a medical facility, the infection is classified as HA-CDI. On the other hand, CA-CDI are infections in non-hospitalized patients in the past 12 weeks and/or symptoms of infection that occurred less than 48 hours from admission to the hospital [3]. Hypervirulent strain NAP1 (North American Pulsed Field Type 1), also known as B1/NAP1/027 or ribotype PCR 027, caused an increase in morbidity and mortality in Europe and throughout the world [4]. Strain 027 has increased resistance to antibiotics, and it produces more spores and toxins A and B and the binary toxin [5]. It should be noted that, regardless of the origin of the strain (HA-CDI or CA-CDI), it is the most commonly encountered (22.9%) ribotype in Europe [6] and in Poland [7].

Antibiotic stewardship limits exposure to CDI in hospitals; hence, according to Barlam et al., one of the elements of the *Antibiotic Stewardship Program* is, on the one hand, application of data on CDI incidence as a measure of exposure to CDI, and on the other hand, monitoring antibiotic use as measured by the Defined Daily Dose (DDD)—a technical unit of measurement of drug consumption, defined by the World Health Organization (WHO) [8,9]. Many of the literature reports, e.g., from Great Britain [10], confirm the above thesis.

Indicators to measure the quality of antimicrobial stewardship in primary care and, secondly, in hospitals were developed and validated by the European Surveillance of Antimicrobial Consumption program [11]. One of them is the ratio of the consumption of broad-spectrum antibiotics: combination of penicillins, including beta-lactamase inhibitor, second- and third-generation cephalosporins, lincosamides and streptogramins (J01(CR+DC+DD+(F-FA01))) to the consumption of narrow-spectrum antibiotic: beta-lactamase-sensitive penicillins, first-generation cephalosporins and macrolides (J01(CE+DB+FA01)).

The objective of the study was to analyze the possibility of utilizing aggregate data concerning CDI incidence and antibiotic consumption at the national level, for the purpose of identifying areas that require intervention at the level of public health.

## 2. Materials and Methods

The statistical analysis was prepared, using publicly available data from the European Centre for Disease Prevention and Control for 2016:

1. Concerning antibiotic consumption based on:

a. Antimicrobial consumption database (ESAC-Net), fluoroquinolones, beta lactams, macrolides, lincosamides and streptogramins expressed as DDD per 1000 inhabitants and per day were taken into consideration, the DDD is defined as the assumed average maintenance dose per day for a drug used for its main indication in adults.

b. The Quality indicator for antibiotic consumption in the community according to ECDC methodology was defined as the ratio of the consumption of broad-spectrum antibiotic (J01(CR+DC+DD+(F-FA01))) to the consumption of narrow-spectrum antibiotic (J01(CE+DB+FA01)); consumption of antibiotics was compared for hospitals and outpatient care; data from The European Surveillance System—TESSy, provided by (Austria, Belgium, Croatia, Estonia, Finland, France, Greece, Hungary, Ireland, Italy, Latvia, Lithuania, Malta, Netherlands, Poland, Slovenia, Spain and UK–Scotland) and released by ECDC (https://ecdc.europa.eu/en/antimicrobial-consumption/database/rates-country, https://www.ecdc.europa.eu/en/antimicrobial-consumption/database/qualityindicators).

2. concerning CDI epidemiology based on surveillance data collected by the ECDC Healthcare-Associated Infections surveillance Network (HAI-Net), in 71.5% hospital surveillance periods, the reported diagnostic practices followed ESCMID recommendations [6].

Quantitative evaluation of the measured variables employed basic statistical measures, i.e., mean, standard deviation, and minimum and maximum values. In order to verify the statistical significance of relationships between antibiotic consumption in ambulatory and hospital care expressed in DDD (defined daily dose) per 1000 inhabitants and per day and CDI incidence and/or incidence density in 2016, Pearson’s correlation coefficients were used. The selection of parametric significance tests was dictated by the fact that the measured variables have probability density distributions converging to the Gaussian distribution. In order to determine the predictive model with 95% confidence, the least squares model was developed. These models were done for variables exhibiting significant correlations, i.e., community and total consumption of antibacterials (Anatomical Therapeutic Chemical, ATC group J01) for systemic use in the hospital/community sector and consumption of other beta-lactam antibacterials (ATC group J01D) for systemic use in the hospital/community sector. All statistical tests were calculated at 95% statistical confidence interval, i.e., at the level of significance of alpha = 0.05.

## 3. Results

The mean antibiotic consumption involving in-hospital and ambulatory treatment amounted to 20.4 DDD (SD 7.89, range 11.04–39.69); outpatient care employed 10 times more antibiotics than hospitals: 18.6 DDD vs. 1.9 DDD. Discrepancies between in-hospital and ambulatory antibiotic usage were 9.8 times higher: 18.6 DDD vs. 1.9 DDD, the discrepancy was particularly high for lincosamides and streptogramins (ATC group J01F): 20.3 times more: 3.04 DDD (ambulatory) vs. 0.15 DDD (hospital) (Table 1). This observation does not apply CDI incidence density rate in community-associated and healthcare-associated CDI, in community is 2.9 times smaller, respectively: 0.81 and 2.38.

In total, 44.9% of antibiotics used in outpatient care were broad-spectrum antibiotics. A significant relationship was found between the quality of antibiotics prescribed and their consumption (*p* < 0.001). The more broad-spectrum antibiotics, other beta-lactams (ATC group J01D) for systemic use prescribed, the higher the sales of antibiotics both in total (r = 0.746; *p* < 0.01) and in community sector (r = 0.782; *p* < 0.01) (Figure 1).

Mean CDI incidence density was 4.6 cases/10,000 patient-days (pds), higher in hospital sector than in community: 3.7 vs. 0.9. CDI incidence, both in total and divided into HA-CDI vs. CA-CDI, did not statistically significantly correlate with the remaining factors analyzed. 

There is no significant relationship between the antibiotics consumption and CDI incidence rate. Though not significant, there is near-significant association (*p* = 0.053) between hospital use of “other beta-lactams” and HA-CDI—as expected, this class includes carbapenems, which are the single highest group of antibiotics conferring CDI risk (Table 2).

In Poland, the proportion of antibiotics in outpatient settings was 16.3 times higher than in the hospital environment, and it was one of the highest among the countries studied. Despite the lack of a trend in international analysis, in Poland, very high antibiotic consumption was found: 27.0 DDD in community and 1.79 DDD in hospital sector, as well as increased CDI incidence—one of the highest among the countries studied (7.6/10,000 pds) (Appendix A).

## 4. Discussion

One of the strongest CDI predictors is the use of antibiotics. This is confirmed by the present data in relation to Poland, in which both antibiotic consumption and CDI incidence were among the highest and the latter was more than two times higher than the average for Europe: 7.6 vs. 3.2/10,000 pds. Unfortunately, in Poland, in 2007–2016, outpatient care saw a significant increase in the total sales of antibiotics, by 8.3%; it was also true for the consumption of the group of antibiotics whose relationship with CDI is the strongest, i.e., fluoroquinolones, by 13.5% [13]. The growing consumption of fluoroquinolones is very worrying, especially in view of the British reports that clearly show the necessary direction of change—they associate restricting fluoroquinolone prescribing with the decline in CDI incidence. Antimicrobial stewardship should be a central component of CDI control programs [10] in hospital and in the ambulatory sector.

In hospital care, the departments with particularly high antibiotic consumption are intensive care units (ICUs). A single-center study from a Polish ICU, in which the consumption of antibiotics amounted to a very high level of 1913 DDD/1000 pds, with an alarming CDI incidence, 10.6/10,000 pds, and additionally a simultaneous exceptionally high and disturbing prevalence of high-resistance strains was found [14]. However, the above example was not an exception: A multi-center study in Polish ICUs indicated consumption of antibiotics at an average level of 1520 DDD/1000 pds (ranging from 620 to 3960 DDD/1000 pds). These data were significantly higher than analogous rates recorded in, e.g., Germany, 1305 (463–2216), or Sweden, 1147 (605–2134 DDD/1000 pds) [15,16,17]. What is also worrying in Polish ICUs is the high contribution of carbapenems (17%) and quinolones (14%, the second antibiotic class) that was found, while it is precisely the proportion of broad-spectrum antibiotics and fluoroquinolones in the total antibiotic consumption that constitutes a parameter evaluating the quality of prescribing antibiotics [17].

Other departments in which CDI poses a problem are surgical wards, where, on the one hand, perioperative antibiotic prophylaxis is applied, including, e.g., fluoroquinolones in urology, and in the treatment of healthcare-associated infections, including surgical site infections, broad-spectrum antibiotics are used [18]. According to Abdelsattar et al., the incidence (microbiologically confirmed CDI) in patients after general, vascular or gynecological surgery in the USA was estimated at 0.1–2.6%, depending on the type of the procedure [19]. The incidence of CDI among Polish patients after neurosurgical operations was similar: 0.1% [20].

One of the elements of a comprehensive program of antibiotic management at the hospital level is the evaluation of their consumption and cost analysis covering both the costs of purchasing and administering the antibiotic and the remaining infection treatment costs [21,22]. Many studies confirm the relationship between the effective implementation of the hospital antibiotic policy, a reduction in antibiotic consumption and a decrease in CDI incidence [8,16,23], which was not corroborated through analysis of aggregate data at the European level. However, the method in these studies varies—especially the measurement of the consumption of antibiotics. In the majority of countries, ECDC summary reports are based on sales data and the DDD is referred to 1000 people per day. As demonstrated in research at the hospital level, pharmacy data concerning the issuing of antibiotics differ significantly from those presenting actual use [22,24]. Probably, a better denominator would be a narrower population (e.g., person-days of hospitalization or the number of admissions to a particular ward), as is the case in many HAI surveillance programs, in which the so-called target surveillance is carried out. It is likely that methodological approach plays a key role here and perhaps this is the reason behind the fact that the analysis of data presented in ECDC reports on antibiotic consumption and CDI incidence showed no significant relationships. Moreover, there is likely to be variability in CDI diagnosis among European countries that can greatly affect CDI incidence rates or detection of CDI. According to the ECDC report, only 71% surveillance were prepared by using the full ESCMID-recommended diagnostic algorithms. Meanwhile 29% surveillance performed CDI diagnosis with GDH (glutamate dehydrogenase), confirmation with NAAT (Nucleic acid test) or GDH, confirmation with toxigenic culture, NAAT alone, or with toxin detection, confirmation with NAAT or toxigenic culture, toxigenic culture alone, EIA for toxins alone, stool cytotoxicity assay alone [6].

Regardless of the problem of research methodology limitations indicated in this study, the introduction of a sensible antibiotic policy in many of the countries studied seems to be a necessity. For Polish hospitals, it might prove to be a considerable challenge. Poland, as one of the Central and Eastern European countries, represents an example of a poorly functioning epidemiological surveillance system, which was demonstrated by Ider et al. on the example of healthcare-associated infection (HAI) surveillance, since in the countries of the post-Soviet bloc, there is a problem of weak commitment and insufficient resources, especially regarding the staff, lack of specialist knowledge and unsatisfactory publication of data on HAI epidemiology [25].

The present data indicate that interventions concerning surveillance of antibiotic consumption have to encompass not only hospitals but also physicians in outpatient care, since a significant portion of the antibiotic market involves ambulatory patients [26]; in Poland, it was over 16 time greater. In the Polish outpatient treatment conditions, two independent aspects will probably be a major challenge as regards antibiotic policy: poor access to rapid diagnostic tests (cassette or test strips) or full microbiological diagnostics and to basic health care. In Poland, the number of physicians per 1000 inhabitants is the lowest in Europe and amounts to 2.3, with the European average of 3.6 [27], which probably contributes directly to both increased consumption of antibiotics and to their irrational prescribing. It should be emphasized that the sale of antibiotics in the community is much higher than in hospitals: It was 16 times greater. Moreover, the results presented significantly point to another vital element necessary in the implementation of rational antimicrobial stewardship in ambulatory care: The problem is not only the total use of antibiotics, but in particular, the application of broad-spectrum antibiotics, since the elevated percentage concerning this group of antimicrobials is closely correlated with high total sales of antibiotics. Hence, it seems particularly important to enhance not only the activities around appropriate antibiotic prescribing and use in the pre- and post-graduate training of physicians, but also to introduce limitations in antibiotic prescribing in ambulatory care, e.g., a basic list of medicines with reference prices and a supplementary list which includes more expensive medicines or other solutions.

## 5. Limitations

The 2016 CDI data of HAI-Net are not fully representative; the collection of incidence data on CDI at European level started on 1 January 2016. Several countries have not submitted data at all (e.g., Germany and UK–England) or data from 1–4 hospitals only (e.g., Austria, Greece, Italy, Latvia, Netherlands and Spain). Some countries have submitted data from a larger number of hospitals (e.g., France, Belgium, Poland, Hungary and Slovakia). The ESAC-Net, the source of data on antibiotic consumption, has been running for many years now; it is a well-established surveillance. Due to the above issue, no valid conclusions can be derived directly from the comparative results of the study (antibiotic consumption vs. CDI incidence).

## 6. Conclusions

Antibiotic consumption and the CDI incidence may depend on many national variables associated with local systems of healthcare organization and financing. Their interpretation in international comparisons requires caution. It is also a confirmation that looking for solutions to problems as universal as CDI prevention and control necessitates detailed descriptions of the local situation and taking into account local conditions and limitations. Aggregate data concerning antibiotic consumption at the national level do not lend themselves to detailed analyses as regards CDI.

## Figures and Tables

**Figure 1 antibiotics-09-00127-f001:**
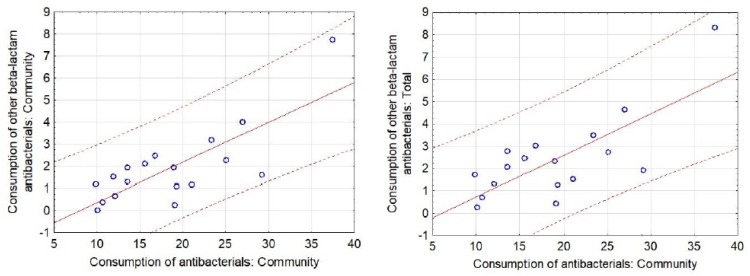
Scatter diagram with 95% prediction curve for significant correlations between the respective categories of Community and Total Consumption of antibacterials (ATC group J01) for systemic use in the hospital/community sector and Consumption of other beta-lactam antibacterials (ATC group J01D) for systemic use in the hospital/community sector for all of the included European countries.

**Table 1 antibiotics-09-00127-t001:** Descriptive statistics of the measured variables: consumption of antibiotics in outpatient and hospital care expressed in DDD * per 1000 inhabitants and per day and CDI incidence density in 2016 based on [6,12] for all of the included European countries.

Antimicrobial Consumption and CDI Epidemiology		Mean (SD)	Range (min; max)
Consumption of antibacterials (ATC group J01) for systemic use in the hospital/community sector	Hospital	1.9 (0.51)	(0.9; 3.1)
	Community	18.6 (7.55)	(9.9; 37.4)
	Total	20.4 (7.89)	(11.0; 39.7)
Consumption of quinolones (ATC group J01M) for systemic use in the hospital/community sector	Hospital	0.2 (0.08)	(0.1; 0.4)
	Community	1.5 (0.78)	(0.5; 2.8)
	Total	1.7 (0.84)	(0.6; 3.2)
Consumption of other beta-lactam antibacterials (ATC group J01D) for systemic use in the hospital/community sector	Hospital	0.5 (0.20)	(0.2; 0.8)
	Community	1.9 (1.76)	(0.0; 7.7)
	Total	2.4 (1.85)	(0.3; 8.3)
Consumption of macrolides, lincosamides and streptogramins (ATC group J01F) for systemic use in the hospital/community sector	Hospital	0.2 (0.07)	(0.1; 0.3)
	Community	3.0 (1.42)	(0.8; 6.9)
	Total	3.2 (1.44)	(0.9; 7.1)
Quality indicators for antibiotic consumption in the community (primary care sector) **	Broad/narrow	44.90 (57.17)	(0.5; 208.5)
Mean CDI incidence density (per 10,000 pds)	Hospital	3.7 (2.80)	(1.6; 12.9)
	Community	0.9 (0.45)	(0.0; 1.9)
	Total	4.6 (3.05)	(2.5; 14.8)

* DDD: defined daily dose; ** defined as the ratio of the consumption of broad-spectrum (J01(CR+DC+DD+(F-FA01))) to the consumption of narrow-spectrum penicillins, cephalosporins and macrolides (J01(CE+DB+FA01)); CDI, *Clostridioides difficile* infection; pds patient-days; SD, standard deviation.

**Table 2 antibiotics-09-00127-t002:** Pearson correlation coefficient matrix 2 in analysis: consumption of antibiotics in outpatient and hospital care expressed in DDD * per 1000 inhabitants and per day and CDI incidence density in 2016, based on [12] for all of the included European countries.

Quality Indicator for Antibiotic Consumption		Mean CDI Incidence Density		
		HA-CDI	CA-CDI or Unknown Origin	Total
Consumption of antibacterials (ATC group J01) for systemic use in the hospital/community sector	Hospital	r = −0.112	r = 0.029	r = −0.099
		*p* = 0.668	*p* = 0.911	*p* = 0.706
	Community	r = −0.283	r = 0.356	r = −0.207
		*p* = 0.271	*p* = 0.161	*p* = 0.426
	Total	r = −0.281	r = 0.347	r = −0.207
		*p* = 0.274	*p* = 0.173	*p* = 0.426
Consumption of quinolones (ATC group J01M) for systemic use in the hospital/community sector	Hospital	r = −0.043	r = −0.052	r = −0.047
		*p* = 0.871	*p* = 0.842	*p* = 0.859
	Community	r = −0.31	r = 0.016	r = −0.28
		*p* = 0.226	*p* = 0.953	*p* = 0.276
	Total	r = −0.292	r = 0.01	r = −0.265
		*p* = 0.255	*p* = 0.970	*p* = 0.303
Consumption of other beta-lactam antibacterials (ATC group J01D) for systemic use in the hospital/community sector	Hospital	r = 0.477	r = 0.042	r = 0.442
		*p* = 0.053	*p* = 0.873	*p* = 0.076
	Community	r = −0.015	r = 0.241	r = 0.022
		*p* = 0.956	*p* = 0.352	*p* = 0.935
	Total	r = 0.036	r = 0.236	r = 0.067
		*p* = 0.890	*p* = 0.363	*p* = 0.799
Consumption of macrolides, lincosamides and streptogramins (ATC group J01F) for systemic use in the hospital/community sector	Hospital	r = −0.23	r = −0.108	r = −0.226
		*p* = 0.375	*p* = 0.681	*p* = 0.383
	Community	r = −0.153	r = 0.241	r = −0.106
		*p* = 0.557	*p* = 0.352	*p* = 0.687
	Total	r = −0.162	r = 0.232	r = −0.115
		*p* = 0.534	*p* = 0.371	*p* = 0.660
Quality indicators for antibiotic consumption in the community (primary care sector) **	Broad/narrow	r = −0.278	r = −0.096	r = −0.268
		*p* = 0.279	*p* = 0.715	*p* = 0.298

* DDD: defined daily dose; ** defined as the ratio of the consumption of broad-spectrum (J01(CR+DC+DD+(F-FA01))) to the consumption of narrow-spectrum penicillins, cephalosporins and macrolides (J01(CE+DB+FA01)). CA community-associated; CDI, *Clostridioides difficile* infection; HA healthcare-associated; SD, standard deviation.

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
