# Peer review of "Consumption of Antibiotics and Epidemiology of Clostridioides difficile in the European Union in 2016—Opportunity for Practical Application of Aggregate ECDC Data"

_antibiotics, 2020, doi:10.3390/antibiotics9030127_

Round 1
Reviewer 1 Report
While this manuscript by Jachowicz et al. contains very interesting results, I believe it needs some textual revision. The results section was perfunctory and needs to be expanded upon. A more comprehensive write-up of the information presented in Tables 1 and 2 would be helpful. For example, I was confused about the ‘broad/narrow’ quality indicator in the last row of Table 2 and would have appreciated elaboration. Moreover, Tables 1 and 2 appear to contain average data from several European countries, but the discussion focuses on data from Poland, which is not listed in any figure or table. An additional table containing the same information about Poland specifically (or several supplemental tables containing the country-specific information for all countries analyzed) would be appropriate.
Specific Notes:
I would like to see a better definition of the defined daily dose. The one given by the World Health Organization (Defined Daily Dose (DDD): The assumed average maintenance dose per day for a drug used for its main indication in adults) would suffice.
Please, please reformat the graphs in graphs in Figure 1 to have the axis labels in a readable font size. I had to blow the file up to 250% magnification to read them.
Consider presenting the data in Table 1 as a bar graph. The finding that outcare antibiotic sales and CDI rates were so much higher than those in hospitals was striking, and I feel it was underemphasized in the text. Note: Table 1, Translate Srednia to Mean
Table 2, consider putting the statistically significant correlations in bold text to highlight the relevant relationships.
Titles and legends Tables 1 and 2 do not clearly state whether the numbers listed are for all of the European countries as aggregate values or are specific to Poland. I believe they are the pan-Europe average, but this was not clear as presented.
Assuming that Table 2 lists average values across Europe, given the textual references to the Poland-specific data, consider adding a Table 3 with the Polish data.
Author Response
DETAILED RESPONSE TO REVIEWERS:
STEP-BY-STEP REPLIES TOREVIEWERS' COMMENTS:
While this manuscript by Jachowicz et al. contains very interesting results, I believe it needs some textual revision.
Authors’ reply: Thank you for this comment!
The results section was perfunctory and needs to be expanded upon. A more comprehensive write-up of the information presented in Tables 1 and 2 would be helpful. For example, I was confused about the ‘broad/narrow’ quality indicator in the last row of Table 2 and would have appreciated elaboration. Moreover, Tables 1 and 2 appear to contain average data from several European countries, but the discussion focuses on data from Poland, which is not listed in any figure or table. An additional table containing the same information about Poland specifically (or several supplemental tables containing the country-specific information for all countries analyzed) would be appropriate.
Specific Notes:
I would like to see a better definition of the defined daily dose. The one given by the World Health Organization (Defined Daily Dose (DDD): The assumed average maintenance dose per day for a drug used for its main indication in adults) would suffice.
Authors’ reply: Corrected according to suggestions, as below:
Please, please reformat the graphs in graphs in Figure 1 to have the axis labels in a readable font size. I had to blow the file up to 250% magnification to read them.
Authors’ reply: Corrected according to suggestions, sorry and thank you for your patience.
Consider presenting the data in Table 1 as a bar graph.
Authors’ reply: Table 1 presents a large number of data that are expressed in various scales and in very different values, from 0.5 DDD (Consumption of other beta-lactam antibacterials (ATC group J01D) for systemic use in the hospital sector) to 44.90 (Quality indicators for antibiotic consumption in the community). In our opinion, Table 1 provides a more consistent and qualitative picture of the values of variables that are on a different scale.
The finding that outcare antibiotic sales and CDI rates were so much higher than those in hospitals was striking, and I feel it was underemphasized in the text.
Authors’ reply: Corrected according to suggestions, as below:
“(…) Discrepancies between in-hospital and ambulatory antibiotic usage were 9.8 times higher: 18.6 DDD vs. 1.9 DDD, the discrepancy was particularly high for lincosamides and streptogramins (ATC group J01F): 20,3 times more: 3.04 DDD (ambulatory) vs. 0.15 DDD (hospital) (Table 1). This observation does not apply CDI incidence density rate in community-associated and healthcare-associated CDI, in community is 2.9 times smaller, respectively: 0.81 and 2.38. (…)”
Note: Table 1, Translate Srednia to Mean
Authors’ reply: Corrected according to suggestions, sorry and thank you for your patience.
Table 2, consider putting the statistically significant correlations in bold text to highlight the relevant relationships.
Authors’ reply: Unfortunately there no any significant correlations in table 2, except the near-significant association (p=0.053) between hospital use of “other beta-lactams” and HA-CDI.
Titles and legends Tables 1 and 2 do not clearly state whether the numbers listed are for all of the European countries as aggregate values or are specific to Poland. I believe they are the pan-Europe average, but this was not clear as presented.
Authors’ reply: Corrected according to suggestions.
Assuming that Table 2 lists average values across Europe, given the textual references to the Poland-specific data, consider adding a Table 3 with the Polish data.
Authors’ reply: Corrected according to suggestions, the table 3 as supplementary material was added.

Reviewer 2 Report
This manuscript aims to relate national antibiotic use in inpatient and outpatient settings to C. difficile rates (both hospital acquired and community-onset) in the same countries across Europe by merging 2 data sets.
No relation was found and the authors speculate on potential explanations. There is a strong underlying assumption that there SHOULD be a relationship . They cite the example of their own country, Poland, which is an outlier in both high CDI rates and high antibiotic use. They assume that they failed to detect a relationship across the board. Specifically, they state the C Diff data set is not mature and may not be representative of the situation country-wide
Interesting findings unrelated to CDI rates is that outpatient antibiotic use is about 10-foldhigher than inpatient use. Further, the only significant relation found is between use of broad-spectrum beta-lactam antibiotics and overall use of antibiotics in both the outpatient setting, and in all settings .
Comments:
Abstract – Nowhere is there mention of the conclusion related to CDI rates –i.e. no relation of CDI rates to antibiotic use on a country-wide level.
-The “conclusions” comments are speculative, and not actually shown by this study.
Introduction –
- Last paragraph - :”employing utilizing” - eliminate one of these
Materials &Methods-
-IS there a reference or precedent for the defined “Quality indicator” of antibiotic use, relating “Broad-spectrum” to “narrow-spectrum” antiibotics.
-what are the countries used in HAI-net 2017 – at end of the paper, its state Poland is not included in the 2017 dataset.
-define ATC group
Results
-“total antibiotic consumption” should be mean or average antibiotic consumption
- -Suggest 2nd sentence be written as : Discrepancies between in-hospital and ambulatory antibiotic use were particularly high for ….,overall comprising 15.6% of total consumption.
-Mention the lack of association of CDI rates with antibiotic use!!
-Though not significant, there is near-significant association (p=0.053)between hospital use of “other beta-lactams” and HA-CDI – as expected if this class includes carbapenems, which are the single highest group of antibiotics conferring CDI risk.
Discussion
-4th paragraph –…which was not corroborated here through analysis…level. However, the methods in these different studies vary –
- -6th paragraph -..it was 16 time greater.
Conclusions
Should be the last section!
3rd sentence – problems, as universal as CDI (eliminate also)
Limitations
Define AER
There is likely to be variability in CDI diagnosis among European countries that can greatly affect CDI incidence rates , by up to 50%– I am not aware if there is a standard European algorithm for using PCR vs toxin testing for diagnosis, but this introduces huge discrepancies in incidence rates! If there are national guidelines, perhaps PCR-using countries can be analyzed separately form toxin-utilizing countries.
Author Response
DETAILED RESPONSE TO REVIEWERS:
STEP-BY-STEP REPLIES TOREVIEWERS' COMMENTS:
This manuscript aims to relate national antibiotic use in inpatient and outpatient settings to C. difficile rates (both hospital acquired and community-onset) in the same countries across Europe by merging 2 data sets.
No relation was found and the authors speculate on potential explanations. There is a strong underlying assumption that there SHOULD be a relationship . They cite the example of their own country, Poland, which is an outlier in both high CDI rates and high antibiotic use. They assume that they failed to detect a relationship across the board. Specifically, they state the C Diff data set is not mature and may not be representative of the situation country-wide. Interesting findings unrelated to CDI rates is that outpatient antibiotic use is about 10-foldhigher than inpatient use. Further, the only significant relation found is between use of broad-spectrum beta-lactam antibiotics and overall use of antibiotics in both the outpatient setting, and in all settings.
Authors’ reply: Thank you for this comment!
Comments:
Abstract – Nowhere is there mention of the conclusion related to CDI rates –i.e. no relation of CDI rates to antibiotic use on a country-wide level.
Authors’ reply: Corrected according to suggestions.
-The “conclusions” comments are speculative, and not actually shown by this study.
Authors’ reply: Corrected according to suggestions.
Introduction – last paragraph - :”employing utilizing” - eliminate one of these
Authors’ reply: Corrected according to suggestions.
Materials &Methods-
Is there a reference or precedent for the defined “Quality indicator” of antibiotic use, relating “Broad-spectrum” to “narrow-spectrum” antiibotics.
Authors’ reply: Corrected according to suggestions, the “Introduction” section was supplemented, as below:
“(…) Indicators to measure the quality of antimicrobial stewardship in primary care and, the secondly, in hospital were developed and validated by the European Surveillance of Antimicrobial Consumption program [11]. One of them is ratio of the consumption of broad-spectrum antibiotic: combination of penicillins, including beta-lactamase inhibitor, second and third-generation cephalosporins, lincosamides and streptogramins (J01(CR+DC+DD+(F-FA01))) to the consumption of narrow-spectrum antibiotic: beta-lactamase-sensitive penicillins, first-generation cephalosporins and macrolides (J01(CE+DB+FA01)). (…)”
what are the countries used in HAI-net 2017 – at end of the paper, its state Poland is not included in the 2017 dataset.
Authors’ reply: The analysis encompassed data concerning CDI incidence and antibiotic consumption expressed as defined daily doses (DDD) and quality indicators for antimicrobial-consumption involving both ambulatory and hospital patients in 2016, based on 2 ECDC reports – published in 2018 (the first) and 2017 (the second):
- European Centre for Disease Prevention and Control. Clostridium difficile In: ECDC. Annual epidemiological report for 2016. Stockholm; ECDC; 2018. Available from: https://www.ecdc.europa.eu/en/publications-data/healthcare-associated-infections-clostridium-difficile-infections-annual. Available on 01.06.2019
- ECDC: Summary of the latest data on antibiotic consumption in the European Union ESAC-Net surveillance data November 2017 https://www.ecdc.europa.eu/sites/default/files/documents/Final_2017_EAAD_ESAC-Net_Summary-edited%20-%20FINALwith%20erratum.pdf Available on 01.06.2019
The source of misunderstanding was the difference between the date of data collection (2016) and date of data publication (2017 and 2018). Corrected according to suggestions.
define ATC group
Authors’ reply: Corrected according to suggestions.
Results
“total antibiotic consumption” should be mean or average antibiotic consumption
Authors’ reply: Corrected according to suggestions.
Suggest 2nd sentence be written as : Discrepancies between in-hospital and ambulatory antibiotic use were particularly high for …., overall comprising 15.6% of total consumption.
Authors’ reply: Corrected according to suggestions, as below:
“(…) Discrepancies between in-hospital and ambulatory antibiotic usage were 9.8 times higher: 18.6 DDD vs. 1.9 DDD, the discrepancy was particularly high for lincosamides and streptogramins (ATC group J01F): 20,3 times more: 3.04 DDD (ambulatory) vs. 0.15 DDD (hospital) (Table 1). This observation does not apply CDI incidence density rate in community-associated and healthcare-associated CDI, in community is 2.9 times smaller, respectively: 0.81 and 2.38. (…)”
Mention the lack of association of CDI rates with antibiotic use!!
Authors’ reply: Corrected according to suggestions, as below.
“(…) There not significant relationship between the antibiotics consumption and CDI incidence rate. (…)”
Though not significant, there is near-significant association (p=0.053)between hospital use of “other beta-lactams” and HA-CDI – as expected if this class includes carbapenems, which are the single highest group of antibiotics conferring CDI risk.
Authors’ reply: Corrected according to suggestions, as below:
“(…) Though not significant, there is near-significant association (p=0.053) between hospital use of “other beta-lactams” and HA-CDI – as expected, this class includes carbapenems, which are the single highest group of antibiotics conferring CDI risk. (…)”
Discussion
4th paragraph –…which was not corroborated here through analysis…level. However, the methods in these different studies vary –
Authors’ reply: Corrected according to suggestions, as below:
“(…) However, the method in these studies varies (…) Also, there is likely to be variability in CDI diagnosis among European countries that can greatly affect CDI incidence rates or detection of CDI. According to ECDC report only 71% surveillance were prepared using the full ESCMID-recommended diagnostic algorithms. While 29% surveillance performed CDI diagnosis with GDH (glutamate dehydrogenase), confirmation with NAAT (Nucleic acid test) or GDH, confirmation with toxigenic culture, NAAT alone, or with toxin detection, confirmation with NAAT or toxigenic culture, toxigenic culture alone, EIA for toxins alone, stool cytotoxicity assay alone [6]. (…)“
6th paragraph -..it was 16 time greater.
Authors’ reply: Corrected according to suggestions.
Conclusions
Should be the last section!
Authors’ reply: Corrected according to suggestions.
3rd sentence – problems, as universal as CDI (eliminate also)
Authors’ reply: Corrected according to suggestions.
Limitations
Define AER
Authors’ reply: Corrected according to suggestions.
There is likely to be variability in CDI diagnosis among European countries that can greatly affect CDI incidence rates , by up to 50%– I am not aware if there is a standard European algorithm for using PCR vs toxin testing for diagnosis, but this introduces huge discrepancies in incidence rates! If there are national guidelines, perhaps PCR-using countries can be analyzed separately form toxin-utilizing countries.
Authors’ reply: Corrected according to suggestions, as below:
In “Materials and methods” section: “ (…) in 71.5% hospital surveillance periods, the reported diagnostic practices followed ESCMID recommendations [6]. (…)“
In “Discussion” section: “ (…) Also, there is likely to be variability in CDI diagnosis among European countries that can greatly affect CDI incidence rates or detection of CDI. According to ECDC report only 71% surveillance were prepared using the full ESCMID-recommended diagnostic algorithms. While 29% surveillance performed CDI diagnosis with GDH (glutamate dehydrogenase), confirmation with NAAT (Nucleic acid test) or GDH, confirmation with toxigenic culture, NAAT alone, or with toxin detection, confirmation with NAAT or toxigenic culture, toxigenic culture alone, EIA for toxins alone, stool cytotoxicity assay alone [6]. (…)“
